

# The use of pseudo-multivariate standard error to improve the sampling design of coral monitoring programs

Luis M. Montilla, Emy Miyazawa, Alfredo Ascanio, María López-Hernández, Gloria Mariño-Briceño, Zlatka Rebolledo-Sánchez, Andreína Rivera, Daniela S. Mancilla, Alejandra Verde and Aldo Cróquer

Experimental Ecology Laboratory, Universidad Simón Bolívar, Caracas, Miranda, Venezuela

## ABSTRACT

The characteristics of coral reef sampling and monitoring are highly variable, with numbers of units and sampling effort varying from one study to another. Numerous works have been carried out to determine an appropriate effect size through statistical power; however, these were always from a univariate perspective. In this work, we used the pseudo multivariate dissimilarity-based standard error (MultSE) approach to assess the precision of sampling scleractinian coral assemblages in reefs of Venezuela between 2017 and 2018 when using different combinations of number of transects, quadrats and points. For this, the MultSE of 36 sites previously sampled was estimated, using four 30m-transects with 15 photo-quadrats each and 25 random points per quadrat. We obtained that the MultSE was highly variable between sites and is not correlated with the univariate standard error nor with the richness of species. Then, a subset of sites was re-annotated using 100 uniformly distributed points, which allowed the simulation of different numbers of transects per site, quadrats per transect and points per quadrat using resampling techniques. The magnitude of the MultSE stabilized by adding more transects, however, adding more quadrats or points does not improve the estimate. For this case study, the error was reduced by half when using 10 transects, 10 quadrats per transect and 25 points per quadrat. We recommend the use of MultSE in reef monitoring programs, in particular when conducting pilot surveys to optimize the estimation of the community structure.

# INTRODUCTION

The intrinsic value of coral reefs and their relevance in terms of services provided to human societies makes them an object of constant observation; however, because some ecological processes operating in the reefs occur at large spatial and temporal scales, coral scientist face the challenge of obtaining data while keeping a compromise between high precision, reproducibility, and statistical power, with low cost and time (*Aronson et al., 1994*). Several technological advances have allowed a reduction in data variability derived from multiple human observers, e.g., the use of photo- and video-transects instead of *in situ* benthic characterization (*Leujak & Ormond, 2007*); the use of ROV's instead of divers

Corresponding authors
Luis M. Montilla,
luismmontilla@usb.ve
Aldo Cróquer, acroquer@usb.ve

(*Lam et al., 2006*); or the use of artificial intelligence to assist the annotation of photo-quadrats (*Beijbom et al., 2015*; *Williams et al., 2019*; *González-Rivero et al., 2016*).

Coral reef typical sampling and monitoring have usually relied on plot methods, like belt transects; plotless methods, like line-intercept approach; or a combination like the point-intercept approach. The choice and the sampling effort have varied from study to study or from program to program, e.g., *Aronson et al. (1994)* proposed ten 25 m transects with about 50 quadrats containing 10 random points, for sampling spur-and-groove habitats. The CARICOMP protocol relied on the chain method to identify the substrate underneath each chain link on ten 10 m transects (*CARICOMP, 2001*); while AGRRA protocol gets estimates from six 10 m transects for shallow reefs (*Lang et al., 2010*). All these methods have had different amounts of popularity, cost-benefit ratios, and levels of precision associated to the estimation of coral cover (*Nadon & Stirling, 2006*; *Leujak & Ormond, 2007*).

Statistical power has received particular attention in this matter as a tool to decide when a set of conditions allowed the researchers to detect appropriate effect sizes (*Aronson et al., 1994*; *Brown et al., 2004*; *Lam et al., 2006*; *Leujak & Ormond, 2007*; *Molloy et al., 2013*; *Houk & Van Woesik, 2006*), which typically have been studied as univariate analysis of total or mean coral cover (or other particular substrates). This kind of criteria are used even if the research question is related to multivariate cases, which are particularly relevant because it is also important to understand changes occurring not only in the cover of the substrate, but also in the assemblage and functional structure of corals and other reef organisms (*Wulff, 2001*; *Alvarez-Filip et al., 2013*).

*Anderson & Santana-Garcon (2015)* proposed a multivariate approach for estimating the precision of sampling when it is of interest to perform a dissimilarity-based multivariate analysis: the **pseudo multivariate dissimilarity-based standard error (MultSE)**.

In this study we pretend to answer two questions: (1) What values of multSE are achieved in surveys of scleractinian coral assemblage conducted in 36 sites of Venezuela? After determining the reference values, (2) what would be the effect over the multSE of using different combinations of sampling strategies, including variable number of points, quadrats, and transects? This information will help to determine the best sampling protocol and potentially improve the precision in future samplings of these communities.

## METHODS

We used a dataset from 36 previously surveyed sites on seven localities of Venezuela between 2017 and 2018 (detailed in Supp 1) In order to increase the number of sampling sites, we used a variation of the Global Coral Reef Monitoring Network protocol (*2016*) and used four instead of five 30 m transects, 15 photo-quadrats per transect, placed one another meter, and 25 random points per quadrat. The photo-quadrats were annotated using Photoquad (*Trygonis & Sini, 2012*); this generated a matrix of 143 observations × 43 species.

We calculated the MultSE using the scripts available as supplementary material in *Anderson & Santana-Garcon (2015)*, based on a quantitative Bray–Curtis dissimilarity

matrix. Next, we chose the 12 sites with the highest interquantile range (5%–95%) of the MultSE and re-annotated the transects using 100 uniformly distributed points. This allowed us to create a new data set using resampling techniques accounting for different number of transects per site, quadrats per transect and points per quadrat.

### Resampled datasets

Resampling techniques allowed us to create new data sets containing up to 20 simulated transects per site, with 5 to 25 quadrats per transect, and evaluating 25, 50, and 75 points per quadrat. We constructed a script to generate resampled data sets containing transects with all possible combination of number of transects per site, quadrats per transects, and points per quadrats desired. Broadly, from an input specifying the desired parameters:

1. Given the number of desired transects per site ($nbT$), the algorithm lists the available transects and adds repetitions of the same set of transects if $nbT$ is greater than the actual available transects, until $nbT$ is reached; for each simulated transect, the real transects factors and raw CSV file path are extracted.

2. To simulate the desired number of quadrats per simulated transect ($nbQ$), the algorithm reads each transect raw CSV file and, depending on the number of real quadrats ($NQ$) makes a sampling with (if $nbQ > NQ$) or without replacement (if $nbQ \leq NQ$) of the observed quadrats.

3. Finally, for points resampling within each simulated quadrat, the algorithm makes a random sampling without replacement, considering the desired number of points ($nbP$), which could not be higher than the observed number of points per quadrat (100 points per quadrat in this study). The algorithm flowchart can be observed in Fig. 1.

### Statistical analysis

We used a linear regression on Box–Cox-transformed data to test effect of each variable on the MultSE. We performed all the data manipulation and statistical analyses with R (*R Core team, 2019*). All our data, scripts, and supplementary material are available as a research compendium at https://github.com/luismmontilla/coral_muse.

## RESULTS

The MultSE of the field data was variable among sites and locations. Some locations had relatively uniform errors among all their sites, while other cases, like Los Roques and Mochima, included sites with largely different values; additionally, some of the highest multivariate error values had also associated the largest interquartile ranges (Fig. 2). These patterns in general were positively, but lowly correlated to the standard error of the mean coral cover and species richness ($r = 0.14$ and $r = 0.18$ respectively, Fig. 3). The mean value of our field data was $0.16 \pm 0.03$, however this value is purely referential for future studies using this metric under the same conditions.

As expected, the magnitude of the MultSE stabilized with the addition of more transects to the data set, however, analyzing more points per quadrat or using more or fewer quadrats didn't improve considerably the estimation (Fig. 4). The linear regression confirmed this, showing that an increase in the number of transects is the best approach to achieve lower
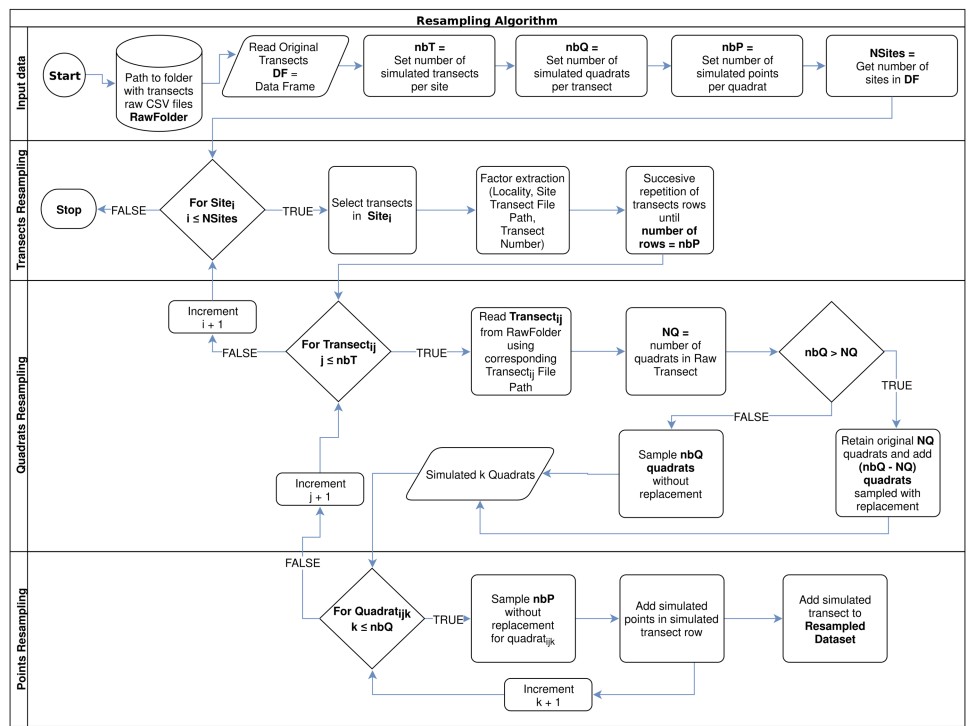

**Figure 1** Resampling algorithm flowchart used to simulate new data per sampled site, according to a desired number of transects per site (***nbT***), number of quadrats per transect (***nbQ***), and number of points per quadrats (***nbP***).

error values ($Estimate = -0.26$, $t = -23.72$, $p = 2 \times 10^{-16}$). Despite all the sources of variation having low $p$-values, the largest reduction in the estimator was observed for transects alone. Unexpectedly, increasing the number of points and quadrats seem to slightly increase the error ($Estimate = 0.029$, $t = 3.47$, $p = 5.26 \times 10^{-4}$, Fig. 5). When selecting a value of ten transects, the MultSE stabilizes near 0.1, representing a reduction of 0.05 units from the average value.

Considering the result of the regression, we used a comparison of ten transects, ten quadrats per transects, and 25 points per quadrat to compare against the original data using the selected resampled sites. In this case, the new error was about half the original estimation in the field (for the subset of sites) using four transects with 15 quadrats per transect (Fig. 6). Furthermore, the inclusion of more transects had an effect on the multivariate ordination of the sites; a principal coordinates analysis with both sampling schemes showed that using ten transects reduced the centroids standard deviation overlap, potentially making easier to discriminate between actual different coral assemblages in these sites (Supp 2).

## DISCUSSION

Here we evaluated the potential of the pseudo multivariate dissimilarity-based standard error as a tool to determine the appropriate number of transects to sample coral assemblages.
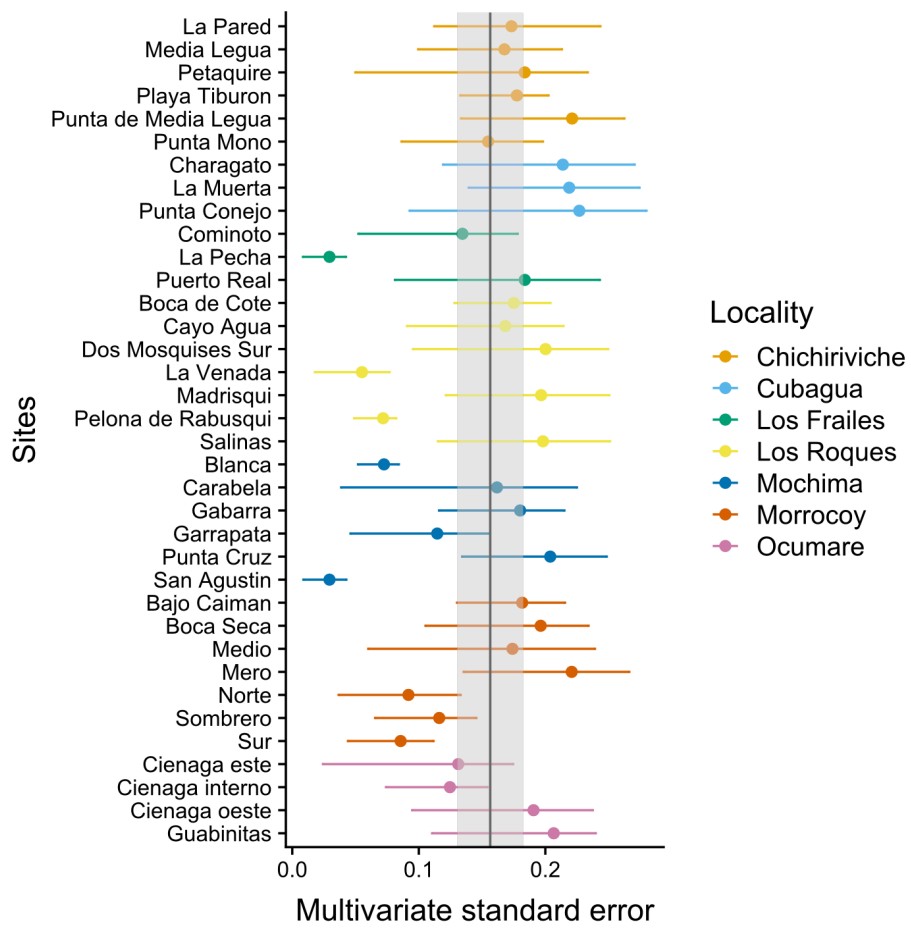

**Figure 2** **Achieved MultSE for each site. Error bars represent 2.5 and 97.5 percentiles.** The vertical band represents the mean ± se.

Though this method is still under development and has its constrains (*Anderson & Santana-Garcon, 2015*), it also provides additional information often overlooked when the research question is related to studying assemblage patterns. We suggest that using at least ten transects instead of four (given that the same dissimilarity measure is used) improves the estimation of the coral assemblages from Venezuela; this number can be partially compensated by reducing the number of quadrats per transect from 15 to 10 quadrats (i.e., 20m transects), and keeping 25 random points per quadrat. To implement this scenario, there are some additional costs that would have to be considered, for example, this would imply increasing the time spent in the field for transect deployment, but this also could be compensated in successive surveys if the researchers use fixed-transects (*Molloy et al., 2013*).

Additionally, for our proposed scheme, the reduction in the number of photoquadrats per transect still results in a net increase of photoquadrats to be analyzed per site, but the impact of this specific step would depend on the size and training of the identification team, the required resolution of the data according to the question, and the tools used to

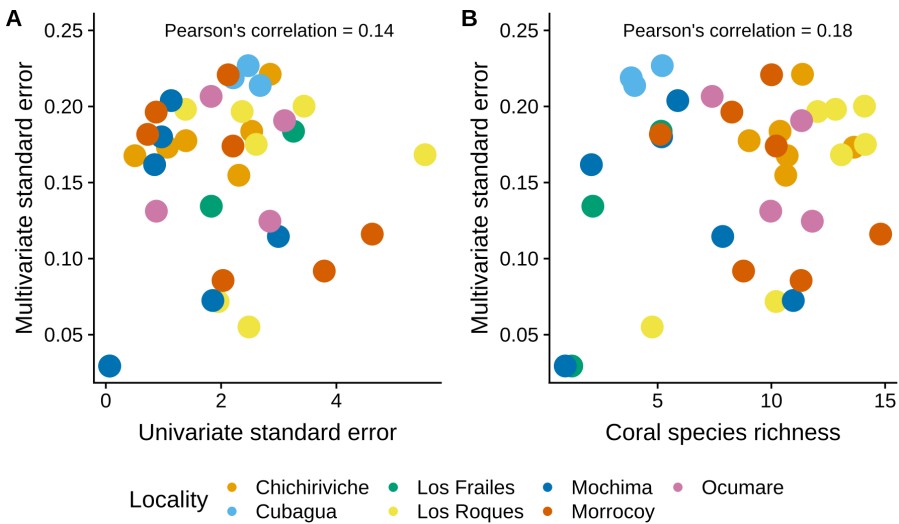

**Figure 3** Comparison of the MultSE with (A) the standard error of mean cover by site, and (B) coral species richness.

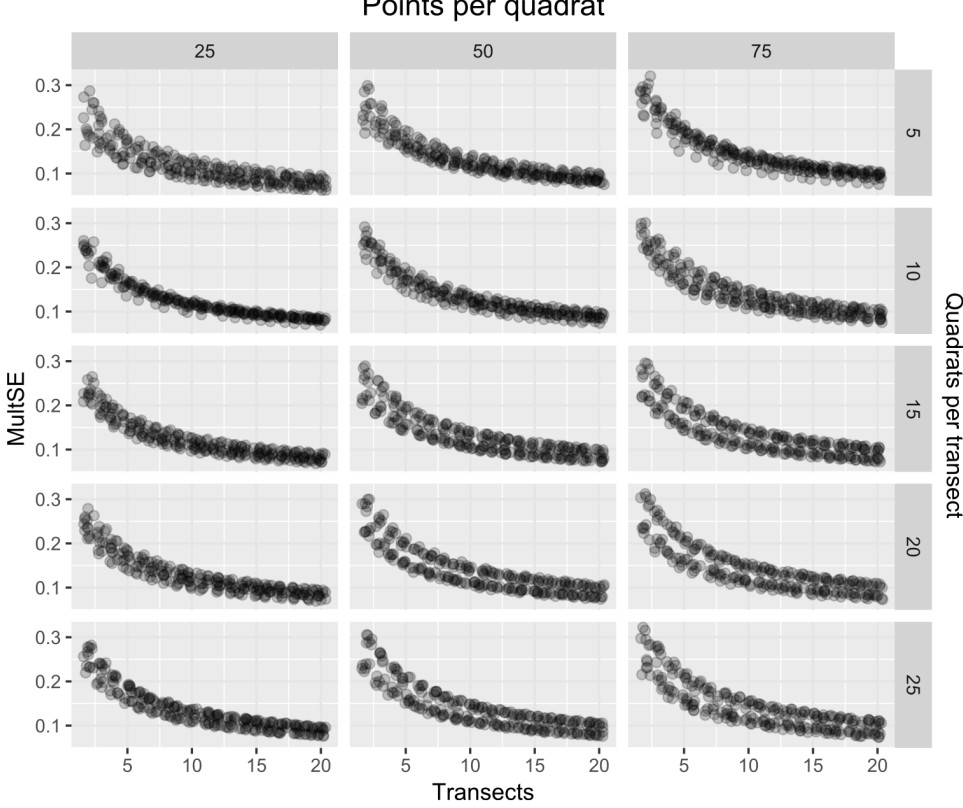

**Figure 4** MultSE for a combination of different number of quadrats (rows), points per quadrat (columns), and transects ($x$-axis).

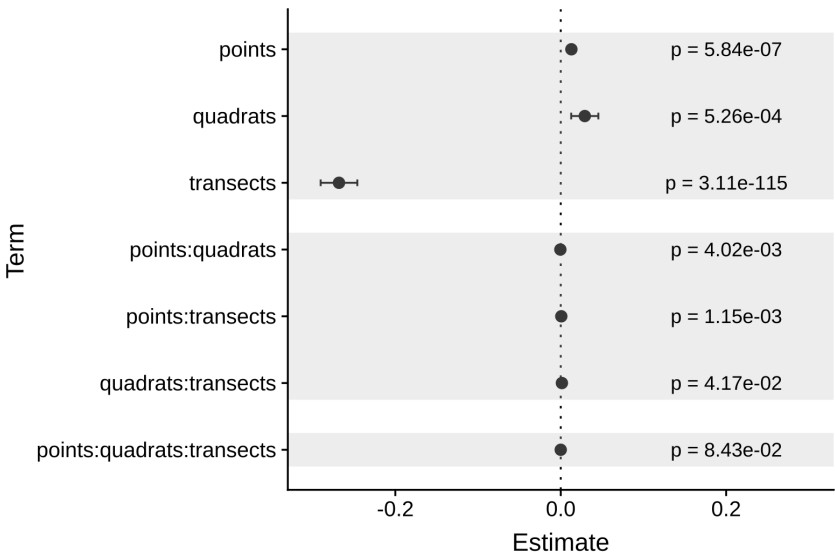

**Figure 5  Coefficients of each source of variation for the linear regression of MultSE.** Negative values imply that an increase of a unit in the respective source of variation, reduces the value of the MultSE.

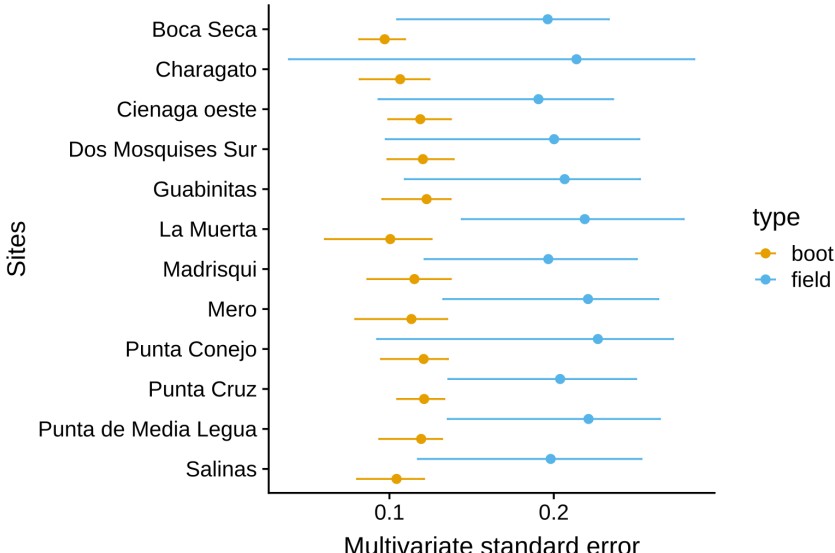

**Figure 6  Comparison of the MultSE for a combination of ten transects, ten quadrats, and 25 random points per quadrat and original sampling scheme.**

generate the dataset. In any case, it is important to highlight that in different locations in would be necessary to implement a similar approach to our proposal here and verify the required number of transects necessary to improve the estimation of the coral assemblage.

Our results also coincide with the findings of *Molloy et al. (2013)*, about the importance of increasing the number of transects instead of quadrats and/or points per quadrat. However, our own comparison indicated that univariate precision of coral cover is

uncorrelated from MultSE, contributing to the idea that this estimation reveals valuable information, specially if the research question is related to compare coral assemblages. We also found that spatial variability had an effect on the MultSE; at the scale of sites that we used —separated by hundred of meters—there were notable differences at the achieved precision, making more necessary the evaluation of the appropriate number of transects to be used. The behaviour of this metric at larger scales still remains to be explored, just as the result on more biodiverse localities e.g., the Indo-Pacific in contrast with the Caribbean.

One example of a practical use of this tool is its incorporation into monitoring programs right at the beginning, assessing the precision of a set of the same number of transects for all the sites; with the results, the researchers can consider redistribute the sampling effort to increase the precision where needed. This can also be performed under an adaptive perspective *sensu* *Lindenmayer & Likens (2009)*; introducing new questions about the coral community would require to take data about coral species instead of only coral cover, keeping the same logistics in the field. In this case, the MultSE can be used to assess the precision of the estimation of the community over time and increase the number of transects, or redistribute the sampling efforts if necessary, and in general, to assess other monitoring tools that rely on multivariate data like multivariate control charts (*Anderson & Thompson, 2004*) can benefit of this approximation.

To conclude, monitoring programs normally put emphasis on fulfilling three pillars: (1) standardization of their sampling protocols, (2) intense training to reduce observer bias and (3) unification of sampling effort. While the practical value of unification is indisputable, we recommend the evaluation of MultSE in coral reef monitoring programs, especially in pilot surveys to optimize the estimation of the coral assemblage across Caribbean geographies.

## ACKNOWLEDGEMENTS

We wish to thank Dr. Rita Peachey and the 39th AMLC scientific meeting organizing committee.

### Funding
This work was funded through a Rapid Ocean Conservation grant from the Waitt Foundation. The funders had no role in study design, data collection and analysis, decision to publish, or preparation of the manuscript.

### Grant Disclosures
The following grant information was disclosed by the authors:
Waitt Foundation.

### Competing Interests
The authors declare there are no competing interests.

## Author Contributions

- Luis M. Montilla conceived and designed the experiments, performed the experiments, analyzed the data, prepared figures and/or tables, authored or reviewed drafts of the paper, and approved the final draft.
- Emy Miyazawa, María López-Hernández, Gloria Mariño-Briceño, Zlatka Rebolledo, Andreína Rivera, Daniela S. Mancilla and Alejandra Verde analyzed the data, authored or reviewed drafts of the paper, and approved the final draft.
- Alfredo Ascanio analyzed the data, authored or reviewed drafts of the paper, performed the algorithm coding, and approved the final draft.
- Aldo Croquer conceived and designed the experiments, authored or reviewed drafts of the paper, and approved the final draft.

## Data Availability

Data is available at GitHub: https://github.com/luismmontilla/coral_muse.

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
