# Peer review of "The use of pseudo-multivariate standard error to improve the sampling design of coral monitoring programs"

_PeerJ, doi:10.7717/peerj.8429_

## Round 0.1 · original submission · Minor Revisions

Three expert reviewers have evaluated your manuscript and their comments can be seen below. As you can see, all reviewers agree that this is an important and well-presented contribution, however, they also have observations and comments that need to be attended to in a resubmission. If you decide to submit a revised version, please ensure that you address all of the points that have been raised.

·

Basic reporting

The explanation of the experiment is clear and the vocabulary seems appropriate.
In the introduction, lines 40 to 45, more description of the methods would be appreciated to explain the differences between each and how this topic of "how many transects are enough" is so important. In method it is said that a variation of the GCRMN method is being used, describe its main characteristics and add citation. More background could be added.
Overall more references would be appreciated in particular in the method section as the whole design seems to be based on the study of Anderson and Santana only.
Figure 2 seems low definition and present a typo at playa tiburon.
Results are very interesting and should be detailed more. Figures are very briefly explained.
Discussion should include more references. Line 131 to 135 are very important points and very interesting for other monitoring teams, more discussion would be relevant. More link to the habitat type or site size should be made.

Experimental design

This research is meaningful and keep adding to the discussion. The experimental design is well described and made available.

Validity of the findings

Results are quickly described and more conclusions could be drawn or described more in depth but they are very interesting and relevant for the line of research.
More interpretation regarding the different monitoring methods stated previously would be appreciated.

·

Basic reporting

The manuscript provides important data regarding the use of different sampling strategies (transects, quadrats and points) to survey coral assemblages. The authors use the multivariate standard error to determine the best sampling protocol in order to get the best estimation of the coral assemblages. An important finding is that increasing the number of transects provided a more precise estimation, while increasing the number of quadrats and points didn’t.

I find the paper to be well written and structured, with only a few minor details included below. The introduction is appropriate and reflects a thorough review of the literature on the topic. The data are clearly presented and interpreted. The discussion is appropriate and supported by adequate references. The conclusions are sound.

In my opinion the study represents a useful contribution to science.

Experimental design

The work is original and the research question is clearly defined, relevant & meaningful.

Validity of the findings

The study is important as it will help to improve the way sampling of coral reefs is done and to have better estimations of the coral assemblages that will allow us to evaluate changes more accurately and to make comparisons between sites and localities.

The authors provide all R codes well structured and described so they can be used to replicate the analysis or to use them for other studies.

Additional comments

ABSTRACT
Ln. 21. It is not clear what authors mean when they say the “re-annotated” a subset of sites. Do they mean “re-sampled”?

METHOD
In the methods section authors only mention that the data came from 36 surveyed sites of Venezuela, however, in the Results Section they also talk about Localities, and this is confusing. I suggest mentioning in this section the number of localities.
Pg. 2. Ln 88. Change “variables” with “variable”

DISCUSSION
Pg. 4 Ln122. Change “spend” to “spent”
Pg. 4 Ln 132. Change “separated at hundred…” to “separated by hundreds”…
Pg. 6 Ln 130 - This sentence is very long, I recommend dividing it in two.

Figure 2. The quality needs to be improved. In Y-axis - Change Playa Tibur?n to Playa Tiburon

Literature
Change: R Core team (2019). R: A language environment for statistical computing. To:
R Core team (2019). R: A language environment for statistical computing. R Foundation for Statistical Computing, Vienna, Austria. https://www.R-project.org/

Reviewer 3 ·

Basic reporting

The article is well written, with appropriate literature for the research topic. Information is available to readers. The figures should improve the resolution of the image, especially Figure 2. The authors should include in the introduction, explicit questions and hypotheses that are evaluated.

Experimental design

The article presents a sampling design and an adequate resampling techniques proposal. However, it must include information on multivariate dataset (nxp) and measures of dissimilarity used.

The authors are interested in assessing the consequences of using different combinations of sample points, quadrants and transects in the multivariate standard error. To evaluate the multivariate standard error, the authors rely on the methodology proposed by Anderson and Santana-Garcon 2015 (included in the literature). However, the statistical proposal of Anderson and Santana-Garcon (2015) refers to the dissimilarity pseudo multivariate-
based standard error (MultSE), measure of precision, not multivariate standard error. An important point is that authors should clarify because in this article they refer to the multivariate standard error and do not use (MultSE). The multivarite standard error and (MultSE), statistically have different implications.This point is key and influences the results obtained, depending on the methodological implications of Anderson and Santana-Garcon (2015)

Validity of the findings

The authors suggest "We suggest that using at least ten transects provides a more precise estimation of the coral assemblages from Venezuela." For this suggestion there are two aspects to consider: 1-How much accuracy in the estimation do the authors refer? and 2-Anderson and Santana-Garcon (2015) suggest that "the value of MultSE in equation (3) (original paper) is dependent on the scale of the dissimilarity measure chosen and (potentially) also on the number of variables, so no such generalization for its value across multiple studies is apparent "; so it influences in the results suggested sample size (least ten transect). Authors should consider reviewing Legendre and De Cáceres Ecology Letters, (2013) 16: 951–963, to select the appropriate dissimilarity measure for dataset (nXp).

The authors suggest "We recommend the evaluation of multivariate standard error in coral reef monitoring programs, especially in pilot surveys to optimize the estimation of the coral assemblage."Two important aspects to consider: 1-What set of variables do the authors refer to when they say "the coral assemblage"? 2- How the authors suggest "estimation of the coral assemblage". These two aspects need to be clarified in the proposed methodology.

Additional comments

This article presents an important topic to consider in environmental and biological monitoring programs. However, there are several aspects (mentioned above) that should be considered by the authors to clarify the methodological approach followed by the authors.

---

## Round 0.2 · Minor Revisions

I have received re-review opinions from three expert reviewers and all have favourable comments, but there are some minor changes that you should consider before I can accept your manuscript for publication. Please address these minor issues in a revised version of your manuscript.

·

Basic reporting

Previous comments were appropriately attended, no further comments

Experimental design

Previous comments were appropriately attended, no further comments

Validity of the findings

Previous comments were appropriately attended, no further comments

·

Basic reporting

No comment

Experimental design

No comment

Validity of the findings

No comment

Additional comments

I have only a few minor suggestions;

Abstract

Pg 5, ln15

I recommend changing:

“In this work, we aimed to assess the pseudo multivariate dissimilarity-based standard error (MultSE) of a series of reefs in Venezuela, sampled between 2017 and 2018, and also, to evaluate the consequences of using different combinations of points, quadrats, and transects over this error.”

To:

“In this work, we use the pseudo multivariate dissimilarity-based standard error (MultSE) approach to assess the precision of sampling scleractinian coral assemblages in reefs of Venezuela between 2017 and 2018 when using different combinations of number of transects, quadrats and points.”


Pg 5, Ln 24:

I recommend changing:

“For this case study, when comparing between sampling with 10 transects, 10 quadrats per transect and 25 points per quadrat; and the original data for Venezuela, we find that the error is reduced by half”

To:
“For this case study, the error was reduced by half when using 10 transects, 10 quadrats per transect and 25 points per quadrat.” (you don’t have to repeat that data for Venezuela are being used).

Introduction

Pg 6, Ln 58
Improve the redaction of last sentence of the introduction:

I recommend changing:

“In this study we wanted to answer what values of multSE were achieved in a series of scleractinian coral assemblage surveys conducted in 36 sites of Venezuela?; and after knowing the reference values, what would be the effect of using different combinations of sampling strategies, including number of points, quadrats, and transects over the multSE? This with the aim of comparing the best sampling protocol and potentially improve precision for future samplings of these communities.”

To:

“In this study we pretend to answer two questions: 1) what values of multSE are achieved in surveys of scleractinian coral assemblage conducted in 36 sites of Venezuela?, and, after determining the reference values, 2) what would be the effect over the multSE of using different combinations of sampling strategies, including variable number of points, quadrats, and transects? This information will help to determine the best sampling protocol and potentially improve the precision in future samplings of these communities


Methods

Pg 6, Ln 64

Change: “We estimated the multivariate standard error”

To: “We estimated the pseudo multivariate dissimilarity-based standard error (MultSE)”

Pg 6, Ln 66

Write what GCRMN means

Change:

“following a variation of the GCRMN monitoring protocol: instead of using five transects per site, we used four 30m transects (to increase the number of sampling sites), 15
photo-quadrats per transect placed one another meter, and 25 random points per quadrat (GCRMN, 2016).

To:

In order to increase the number of sampling sites, we used a variation of the GCRMN monitoring protocol (2016) and used four instead of five 30 m transects, 15 photo-quadrats per transect, placed one another meter, and 25 random points per quadrat.

Pg 10
Fig. 4 needs a title on axis Y2 (number of quadrats)

Reviewer 3 ·

Basic reporting

This section it´s ok.

Experimental design

This section it´s ok.

Validity of the findings

This section it´s ok.

Additional comments

I include two changes that must be added:

Line 64. Change: “We estimated the multivariate standard error (Anderson and Santana-Garcon, 2015)” for “We estimated the MultSE (Anderson and Santana-Garcon, 2015)”.

Line 72. Change to "Matrix of dissimilarity of the Bray-Curtis index". Please, the authors must specify, for the readers, which version of the B-C index they used: quantitative or presence-absence or binary?

---

## Round 0.3 · accepted · Accept

I am satisfied with the changes made to the manuscript.